# Wnt Modulation Enhances Otic Differentiation by Facilitating the Enucleation Process but Develops Unnecessary Cardiac Structures

**DOI:** 10.3390/ijms221910306

**Published:** 2021-09-24

**Authors:** Nathaniel T. Carpena, So-Young Chang, Ji-Eun Choi, Jae Yun Jung, Min Young Lee

**Affiliations:** 1Department of Otolaryngology-Head & Neck Surgery, College of Medicine, Dankook University, 119 Dandae-ro, Cheonan 31116, Korea; ntcarpena@gmail.com (N.T.C.); so4040@hanmail.net (S.-Y.C.); garimung@gmail.com (J.-E.C.); jjkingy2k@gmail.com (J.Y.J.); 2Medical Laser Research Center, Dankook University, Cheonan 31116, Korea; 3Beckman Laser Institute Korea, Dankook University, 119 Dandae-ro, Cheonan 31116, Korea

**Keywords:** otic organoid, wnt pathway, cardiac differentiation

## Abstract

Otic organoids have the potential to resolve current challenges in hearing loss research. The reproduction of the delicate and complex structure of the mammalian cochlea using organoids requires high efficiency and specificity. Recent attempts to strengthen otic organoids have focused on the effects of the Wnt signaling pathway on stem cell differentiation. One important aspect of this is the evaluation of undesirable effects of differentiation after Wnt activation. In the present study, we differentiated mouse embryonic stem cell embryoid bodies (EB) into otic organoids and observed two morphologies with different cell fates. EBs that underwent a core ejection process, or ‘enucleation,’ were similar to previously reported inner ear organoids. Meanwhile, EBs that retained their core demonstrated features characteristic of neural organoids. The application of a Wnt agonist during the maturation phase increased enucleation, as well as otic organoid formation, in turn leading to sensory hair cell-like cell generation. However, with a longer incubation period, Wnt activation also led to EBs with ‘beating’ organoids that exhibited spontaneous movement. This observation emphasizes the necessity of optimizing Wnt enhancement for the differentiation of specific cells, such as those found in the inner ear.

## 1. Introduction

As a recent development in stem cell technology, organoids are in vitro structures similar in morphology and function to certain body organs. They have been used for a variety of purposes, such as screening for drug discovery and uncovering the pathogenesis of certain diseases. For these applications, the organoids of specific target organs must be highly specific and replicable.

The development of an otic organoid, specifically a cochlear organoid, could resolve current problems in hearing loss research, such as the lack of disease modeling and sources of tissue transplantation. Despite the high incidence of sensorineural hearing loss and the growing number of patients with presbycusis [1,2], there is no cure for this loss of function. The main treatment challenge is the inability of cochlear hair cells, which are responsible for mechanoelectrical transduction (i.e., changing a sound input into a neural signal), to regenerate [3]. As an alternative to artificial rehabilitation devices, such as hearing aids and cochlear implants, the development of a highly specific and replicable cochlear organoid could pave the way for new therapeutics for treating hearing loss.

An otic organoid was first introduced by Koehler and Hashino in 2014 [4]. Subsequent observations of electric signals and several epifluorescence markers confirmed the function of sensory hair cells in this new organoid [5]. However, otic organoids have limited efficiency and specificity. With the currently used differentiation protocol, organoids emerge from unknown structures and not every cellular aggregate generates an otic organoid. In addition, for two inner ear organs, i.e., the cochlea and vestibule (important for hearing and balance, respectively), otic lineage differentiation is not well understood. Thus, further exploration is necessary for effective replication of the delicate and complex structure of the mammalian cochlea. Recent attempts to improve otic organoids have focused on the Wnt pathway, because Wnt agonists were found to increase the efficiency of hair cell-like cell production during the otic organoid differentiation process [6].

The Wnt pathway is related to many differentiation and development processes [7,8,9,10,11]. The currently used differentiation protocol has relatively low efficiency and specificity, and unwanted outcomes of the differentiation process must be monitored after Wnt modulation.

In the present study, we used an in vitro differentiation protocol to generate otic organoids and to examine the different cell fates of two morphologies. In addition, we examined the effects of a Wnt agonist applied during the differentiation process. We observed increased hair cell-like cell generation in the Wnt application group. Furthermore, ‘beating’ organoids, which showed spontaneous movement, appeared in the Wnt group. These were analyzed using epifluorescence.

## 2. Results

### 2.1. Development of EBs and Organoids

Our goal was to develop an organoid that resembled the inner ear, i.e., included sensory cells such as hair cells. Following the protocol of Koehler et al. and Dejonge et al. with a few modifications (such as using the hanging drop technique to form EBs) [4], we exposed the cells to key factors at specific time points during the differentiation process (Figure 1A, see the Material and Methods section for details). To generate the EBs, i.e., the cell clusters/aggregates at the beginning of the differentiation process, the hanging drop technique [12] was used until the EBs reached a certain size (day 3), after which they were transferred to differentiation-permissive media. From day 3 to 8, key factors were sequentially applied. On day 9, the EBs were transferred to maturation media with or without the Wnt agonist CHIR99021 (control and (+) CHIR), where they began to develop small protrusions that may have included otic organoids. As shown in Figure 1B, the EBs had no protruding structures on the surface on day 7 (immediately after application of the key factors). However, on day 15 (during maturation), multiple cystic protrusions were observed (arrow). These cystic protrusions eventually sprouted and formed larger cystic structures on day 30.

### 2.2. Two Different Fates of EBs (Sensory Versus Neural Differentiation)

As described in the Introduction section, the differentiation efficiency and specificity of otic organoids vary, and EBs do not all have identical differentiation steps. Some EBs show a typical sequence of morphological changes, characterized by the ejection of a central high-density area. This process has not been described previously. We have termed this process ‘enucleation.’ In detail, at the start of the development process, each EB was composed of a peripheral area with a small cell population and a central area with a large cell population. As the EBs grew, the size of the central area increased while the thickness of the peripheral area was maintained. During the transition between the differentiation and maturation phases (day 8–9), thinning of the peripheral area was observed and a central, high-cellular-density area began to emerge and separate from the EB. This ejection process was complete by around day 12, and the ejected core progressively degenerated. Following this ‘enucleation’ process, the remaining EB was smaller than a conventional one, i.e., an EB that has not undergone ‘core’ enucleation (Figure 2).

After enucleation, multiple vesicles began to form on the peripheral surface of the enucleated EB. These vesicles further developed into cystic organoids (Figure 3A). In contrast, in EBs that did not undergo an enucleation process, the size of the central high-density area progressively increased and the peripheral low-density area gradually disappeared. Eventually, in nonenucleated EBs, the high-density area expanded to form a solid cell-rich structure with no surrounding membranous structures (Figure 3A). To examine the characteristics of nonenucleated EBs, we performed epifluorescence analysis using antibodies related to the neural structure. Nestin is a type IV filament protein expressed in nerve cells. Nestin-positive cells were clustered in the cell-rich structures and sparse GFAP-positive cell, which is most commonly expressed in astrocytes and ependymal cells. We observed no co-expression of these two antibodies (Figure 3B). In certain parts of the nonenucleated EBs, small linear structures spread to nearby Matrigel, as if they were searching for targets. These structures were stained by Tuj1, which is a neuronal marker, and exhibited nerve fiber extension-like movements (Figure 3C). These outcomes suggest that the nonenucleated EBs differentiated into structures with a neural lineage, such as neurons, astrocytes, and ependymal cells, but not into sensory structures.

The Wnt agonist was introduced at the beginning of the maturation phase, on day 9. The rates of enucleation in groups treated and not treated with the Wnt agonist were 74.2% and 45.8%, respectively (Figure 3D). Thus, the Wnt agonist increased the enucleation rate. Taken together with the previous results showing that nonenucleated EBs differentiate into structures with a neural lineage, these data indicate that the application of Wnt agonists could limit neural differentiation and direct EBs to develop into sensory organs.

### 2.3. Differentiation into Inner Ear Hair Cell-Like Cells and the Effects of Wnt Agonist Application

Previous reports have indicated that modulating/enhancing the Wnt pathway could improve the differentiation of otic organoids. Therefore, we attempted to modulate the differentiation process via application of a Wnt agonist (CHIR99021). Indeed, Wnt agonist application increased both the otic vesicle formation rate and number of otic vesicles per EBs (Figure 4B). The rate of organoid formation indicates the number of chances of developing any organoid structure, whether otic, neural, beating, or hair, from all the generated EBs from the hanging drop. EBs that do not form any organoid do not show any changes in morphology and simply break apart or detach from the culture plate. The otic vesicle formation rate indicates the number of otic vesicles among the developed organoid structure. The number of otic vesicles per EB was counted depending on how many protruding otic organoids were produced from a single EB.

In the final stage of the differentiation process, enucleated EBs with otic vesicles and suspected inner ear organoids were fixed and prepared for epifluorescence analysis and real-time PCR. The sectioned EBs contained multiple noncellular spaces surrounded by Sox2-positive cells, and these structures included myosin VIIa-positive cells. These cells were stained for Brn3c, which is an alternative sensory hair cell marker. We observed the co-expression of Brn3c and myosin VIIa, suggesting the possibility of inner ear organoid differentiation. Cells positive for FM1-43, which serves as a marker of mechanoelectrical transduction, were observed, suggesting the presence of functional hair cell-like cells (Figure 5A).

We conducted real-time PCR to quantitatively access the expression of hair cell differentiation-related genes. EBs that differentiated without the Wnt agonist (control) were sacrificed at 7 and 14 days, and EBs that differentiated with the Wnt agonist (CHIR) were sacrificed at 14 days. We then compared gene expression among the EB groups. Among seven genes (*Sox2, E-Cad, Laminin, Pax2, Pax8, Atoh1*, and *Myo7a*), we only observed a statistical difference in expression between EBs with and without the Wnt agonist at 14 days in two genes: *Pax8* and *Myo7a* (Figure 5B). The detailed real-time PCR data are shown in Appendix A. To confirm this result using epifluorescence analysis, myosin VIIa expression was evaluated in Wnt agonist-treated EBs. We observed an increase in myosin VIIa-positive cells compared to EBs that underwent an ordinary differentiation process (control; Figure 5A). Compared to controls, the noncellular cavities in the Wnt agonist-treated EBs were smaller and were surrounded by multiple layers of myosin VIIa-positive cells (Figure 5C). In addition, we observed an increased expression of hair cell-like cell markers in the Wnt agonist-treated EBs. The smaller size of the noncellular cavities in these EBs might be due to the growth of myosin VIIa-positive cell layers. These data clearly indicate that Wnt agonist treatment enhances hair cell-like cell differentiation.

### 2.4. Wnt Agonist Application Facilitates Differentiation into Other Lineages

In the Wnt agonist-treated EBs, structures with unusual morphologies appeared at various stages of differentiation. We had hoped that Wnt activation would only increase sensory progenitor differentiation. However, the change in EB configurations with respect to the cores after CHIR treatment generated different cell fates (Figure 6A). In addition to the otic and neural organoids, EBs with intact cores also generated a beating organoid. For example, on day 27, the EBs had a large, dense cellular body and small protruding membranous structures. At this time point, we observed movement in the peripheral part of the membranous round structures (Figure 6B and Appendix A. The movement was irregular and resembled the contraction of muscular structures. Meanwhile, the growth of cystic morphologies in enucleated EBs into longitudinal conical structures with moderate cell density at the proximal end (Figure 6C) were observed at multiple time points. Linear structures with morphologies that highly resembled hair follicle organoids appeared in the round section of the conical structures. These resembled previously reported hair organoids [13]. 

The difference between intact and enucleated EBs was also apparent several days after CHIR treatment. There was a statistically significant mean size difference of 2.10 ± 0.58 mm^2^ between the intact and enucleated EBs at day 15 until the end of the experiment on day 30 (Figure 6D). The difference in EB size was largely affected by the enucleation process, as shown by the disappearance of the core that can be found in the control and intact EBs (white encircled area). In Figure 6E, 75% formed otic vesicles out of all the EBs that developed organoids, 25% of which came from intact EBs, while 52% were from enucleated EBs. Intact EBs also generated 6% neural and 4% beating organoids, none of which appeared in enucleated EBs. Organoids resembling hair follicles appeared from enucleated EBs, which represented 15% of the total EBs observed. The difference in effect of Wnt activation was also compared by observing the expression of the Wnt target protein LGR5 at the start of maturation (D12), and it was found that the expression was highly elevated in CHIR-treated samples (Figure 6F, yellow dotted rectangle). LGR5 expression could be observed in the control (−) CHIR group, but CHIR-treated EBs showed higher LGR5 expression. Intact EBs showed a high expression of LGR5 only in the outer layer. Cells expressing LGR5 became evenly distributed within the EB after enucleation.

Structures that exhibited movement (Appendix A) were harvested and stained for epifluorescence analysis (Figure 7). We performed epifluorescence analysis using phalloidin to stain F-actin and CTP1 to stain cardiac cells. Both F-actin and CTP1 were observed. Rich clusters of F-actin-positive cells surrounded the empty hollow spaces in the organoids, which appeared to be able to harbor fluid. CTP1 was observed in the cytoplasm of these F-actin-positive cellular structures, as was TAGLN, which reflects actin cross-linking. Thus, the moving/contracting structures resembled cardiac structures, indicating that it may be possible to develop cardiac organoids. We termed these organoids ‘beating’ organoids as they highly resembled cardiac structures. Without application of the Wnt agonist, no beating organoids were observed. However, more than 30% of the Wnt agonist-treated EBs contained these beating organoids, unexpectedly indicating differentiation into cardiac lineage. Thus, while we observed increased differentiation into multiple stacked layers of myosin VIIa-positive cells, the presence of the beating organoids confirmed the initiation of differentiation into other lineages.

## 3. Discussion

We observed two different fates of EBs during the otic organoid differentiation process. When the EBs reached a certain size, some ejected their core (enucleation) and proceeded to differentiate, while others remained un-nucleated and thus became relatively dense organoids. Enucleated EBs were similar to inner ear organoids, as demonstrated in previous reports [6,12]. In contrast, nonenucleated EBs exhibited features characteristic of neural organoids. These results indicate that maintenance of the otic differentiation pathway necessitates limiting the cellular density or volume of the EBs. After application of a Wnt agonist, the rate of enucleation was increased. This outcome, along with enhanced expression of the myosin VIIa gene and an increased number of myosin VIIa-positive cells surrounding the noncellular cavity, suggested that Wnt agonists can enhance otic differentiation by reducing the number of non-otic-differentiated EBs and increasing the number of myosin VIIa-positive cells within each EB. Myosin VIIa-positive cells have been proposed as inner ear sensory hair cell-like cells because they express Brn3c (another inner ear hair cell marker) and FM1-43 (mechanoelectrical transduction marker). However, a more specific classification of cell types (such as cochlear vs. vestibule or inner vs. outer hair cells) is needed. One of the novel findings of the current study was the differentiation into alternative lineages, as indicated by the presence of beating movements in what appear to be cardiac organoids. These were relatively common, i.e., were observed in up to 30% of the EBs exposed to the Wnt agonist. We discuss the relationship between the Wnt pathway and cardiac organoids below. This observation highlights the necessity of optimizing Wnt enhancement for differentiation into specific cells, such as those found in the inner ear. Inner ear organoids developed among the Wnt agonist-treated EBs, which contained additional unknown structures. Thus, enhancement of the Wnt pathway may lead to non-otic lineages and other unwanted outcomes. This issue could be resolved by application of a Wnt agonist to specific progenitor cells. However, this would involve identification of the morphologies of specific pre-differentiated areas within the EBs.

As explained previously, we identified two different fates of EB differentiation: one with and one without an enucleation process. This enucleation process, or ejection of the core or aggregates of the EB, has not been described previously. Although self-guided ‘inside-out’ rearrangements have been reported [14], these differ from the current enucleation process in that the inside-out rearrangements did not result in the division of the structure into two distinct aggregates. According to a previous report, the central area of the EBs remained pluripotent during the initial stage of differentiation [14], indicating that the central area can differentiate further into neural structures. This would affect the overall fate of the EB, as it would eventually differentiate into a neural organoid that could, in theory, provide a source of neural connections for the developing otic organoids. On the other hand, the enucleation removes the largest source of undifferentiated pluripotent cells that could develop into other lineages.

The density and total number of cells in EBs or aggregates may also regulate enucleation. In this study, as a relatively large number of cells were located in the EB core, the number of cells in the EBs may have been sufficient for further differentiation into other lineages. Alternatively, such differentiation might be related to the overall size of the EB. The diameters of the EBs in the current study were larger than those in previous reports [6,14], so some of the factors introduced during differentiation may not have reached the EB core. This could lead to other differentiation processes, or a lack of differentiation, resulting in ejection of the heterogeneous EB core. We also recommend the use of other embryonic or pluripotent stem cell lines to investigate whether a similar enucleation process occurs upon the modulation of the Wnt signaling pathway or perhaps may only be limited to the cell line used in this study.

In looking at the relationship of Wnt signaling and enucleation, a similar occurrence of tissue patterning during development of the embryo happens during gastrulation. Wnt signaling activity coordinates the action of signals during the progression of gastrulation [15]. Ligands from Wnt-producing cells regulate the signaling activity at gastrulation. A mutation on these Wnt ligands fails to initiate gastrulation and the epiblast persistently expresses pluripotency and neural progenitor markers of the embryos.

Here, we adopted protocols that were optimized in a prior report. In that study, optimal concentrations of CHIR were used to generate myosin VIIa-positive cells [14]. However, while we observed an increased number of myosin VIIa-positive cells, we also observed morphological differences compared to the previous study. Specifically, we found multiple layers of myosin VIIa-positive cell lines, as opposed to a single layer. In prior reports, the application of CHIR led to increased differentiation into myosin VIIa-positive cells, although the cell morphologies were similar with versus without CHIR. Thus, while the number of cells increased, the structural organization remained the same. However, in the present study, the structural organization changed such that the myosin VIIa-positive cells surrounded the cavity in stacked layers. Thus, the functional characteristics of this engineered cell (via CHIR application) might be different. Further analysis is necessary to determine the factors underlying this difference. 

Numerous studies have reported cardiac differentiation from EBs or monolayers of stem cells [10,16,17,18,19,20,21,22,23]. These studies, which aimed to uncover the pathology of heart disease [23] and model the human heart to aid the development of therapeutics [16], demonstrated beating structures similar to the beating organoid observed in the present study. The cardiac organoids in prior studies had a round or cystic shape, which was very similar to the beating organoid observed in the present study. Thus, the unexpected beating organoid in the present study was likely a cardiac organoid. A recent publication reported the effectiveness of a Wnt agonist to differentiate cardiac organoids; that protocol had several benefits compared to those used in other studies [16]. That study, along with this report of the accidental generation of cardiac organoids via application of a Wnt agonist, strongly indicates that the Wnt pathway is related to the differentiation of cardiac organoids.

Organoids have provided an opportunity to recreate the structure and physiology of organs. Likewise, diseases and treatments are also possible to investigate using organoids. Our study has shown that the incorrect implementation of a certain treatment can also cause undesirable effects. Compared to the endogenous control of Wnt regulation, the bulk treatment of the Wnt agonist CHIR99021 on an EB also led to the differentiation of other lineages. This indicates that a more targeted delivery of Wnt agonists is necessary to enhance the otic differentiation process. Refining the time points and limiting the delivery of the agonist into a specific part of the EB, such as a putative progenitor area, could be a good approach. Furthermore, the inner ear could benefit more from a more informed effect of stimulating factors that are used if we ever hope to achieve the intricate architecture of the inner ear. Although many issues remain, the use of Wnt agonists is a very promising strategy for differentiation. Indeed, future hearing loss therapies may involve the application of Wnt agonists to the inner ear, to force hidden sensory progenitor cells to differentiate into sensory cells. The mechanism by which Wnt agonist application facilitates the differentiation of both hair cells and other lineages is an important research target, as this process may lead to changes in the target organ.

## 4. Materials and Methods

### 4.1. Cells and Cell Culture

J1 mouse embryonic stem cells (ES-J1; American Type Culture Collection (ATCC), Manassas, VA, USA) were cultured as described by Chang et al. [12]. Briefly, undifferentiated ES-J1 cells were cultured on gelatin-coated plates without feeder cells and were maintained in an embryonic stem cell (ESC) medium, including high-glucose Dulbecco’s modified Eagle’s medium (DMEM; Sigma-Aldrich, St. Louis, MO, USA) supplemented with 15% (*v*/*v*) heat-activated fetal bovine serum (FBS; ATCC, Manassas, VA, USA), 0.1 mM of β-mercaptoethanol (Gibco, Invitrogen, Carlsbad, CA, USA), 0.1 mM of GlutaMAX (Gibco, Carlsbad, CA, USA), 0.1 mM of ESC-qualified nonessential amino acid (NEAA; Welgene, Daegu, Korea), 1% penicillin-streptomycin (PS; ATCC), 1000 U/mL of leukemia inhibitory factor (Millipore, Merck, Burlington, MA, USA), 3 µM of CHIR99021 (Tocris Bioscience, Bristol, UK), and 0.53 µM of PD035901 (Tocris Bioscience, Bristol, UK) at 37 °C in a 5% CO_2_ incubator with optimal humidity. Ectodermal differentiation was initiated using the Glasgow minimum essential medium (GMEM; Gibco, Carlsbad, CA, USA) supplemented with 1.5% knockout serum replacement (Gibco, Carlsbad, CA, USA), 0.1 mM of β-mercaptoethanol, 1 mM of sodium pyruvate (Stem Cell Technologies, Vancouver, BC, Canada), 0.1 mM of NEAA, and 1% PS. The maturation medium consisted of DMEM/F12 (Sigma-Aldrich Co., St. Louis, MO, USA) supplemented with 1% N2 supplement (Gibco, Carlsbad, CA, USA), 1% GlutaMAX, and 0.1% Normocin (InvivoGen, San Diego, CA, USA).

### 4.2. Embryoid Body and Otic Organoid Formation

Embryoid bodies (EBs) were obtained using the hanging drop technique [12], as follows. After dissociating the ES-J1s with TE buffer, drops containing 4 × 10^5^ cells/mL per 30 μL of maintenance medium were inoculated onto a petri-dish cover and cultured at 37 °C for 48 h under the above-described conditions. Then, the collected EBs were washed with 1× phosphate-buffered saline (PBS) and plated in 24-well plates for differentiation into otic-like organoids at 37 °C in a 5% CO_2_ humidified incubator. The Otic differentiation protocol was adapted from Koehler et al. (2014) and DeJonge et al. (2016). Briefly, before the onset of differentiation, the medium was replaced with ectodermal differentiation medium including 2% Matrigel. Then, on the second day of differentiation, nonneural ectoderm was induced by adding 10 ng/mL of recombinant BMP4 (Stemgent, Beltsville, MD, USA) and 1 μM of TGFβi (SB431542, Stemgent, Beltsville, MD, USA) to the cultured cells. On the third day, preplacodal ectoderm was induced by adding 25 ng/mL of FGF-2 (PeproTech, Rocky Hill, NJ, USA) and 1 μM of BMP4i (LDN-193189, Stemgent, Beltsville, MD, USA). The treated cells were cultured for 3 days, and the medium was replaced on day 9 with maturation medium containing 1% Matrigel with or without 3 µM of CHIR99021. After 48 h, a complete change in maturation media without Matrigel and CHIR99021 was performed. Half of the medium was replaced with maturation medium every other day until day 30. The differentiation procedure is illustrated in Figure 1A.

### 4.3. EB Morphological Changes

The morphological characteristics of the formed EBs were observed using a bright-field microscope (CKX53; Olympus, Tokyo, Japan). The fates of the EBs were recorded and their cellular structures were observed from the differentiation phase to the maturation phase (day 1–30). The enucleation and organoid formation rates were assessed according to the image dates. Measurement of the EBs was conducted using DIXI Imaging Solution program (DIXI Optics, Daejeon, Korea).

### 4.4. Characteristics of the EBs Revealed by Epifluorescence Analysis

To examine the characteristics of the EBs, the cells were fixed with 4% paraformaldehyde in 1× PBS overnight at 4 °C and washed three times with 1× cold PBS for 10 min. The cells were cryoprotected by incubating them in 10%, 20%, and 30% (wt/vol) sucrose-PBS solutions for 30 min each and transferred to cryomolds. Then, the sucrose solution was replaced with an optical cutting temperature compound (Tissue-Tek, Torrance, CA, USA). The samples were snap-frozen with dry ice to fashion them into blocks and cut into 5 µm slices using a cryostat microtome (Leica, Wetzlar, Germany). Then, the sectioned samples were mounted onto slides and stained for imaging.

After mounting, the cells were permeabilized with 0.25% Triton X-100 (Sigma-Aldrich Co., St. Louis, MO, USA) in 1 × PBS for 10 min at room temperature (RT), and then blocked for 1 h at RT with 10% normal goat serum (NGS; Vector Laboratories, Burlington, ON, Canada) and 0.1% Triton X-100 to prevent nonspecific binding. To identify nestin-, βIII-tubulin (Tuj1)-, Sox2-, Brn3c-, and myosin VIIa-positive cells inside the formed EBs, the cells were incubated with primary antibody in PBS containing 3% NGS and 0.1% Triton X-100 overnight at 4 °C. The cells were then washed three times with PBS for 5 min each, and incubated with the corresponding secondary antibody for 1 h. Nuclei were visualized using 4′,6-diamidino-2-phenylindole (DAPI). To stain cardiac cells in the organoids, fluorescein phalloidin (F-actin), transgelin (TAGLN)/SM22, and cardiac troponin I (CTP1) were applied using the above protocol. Representative images were acquired using a confocal microscope (Olympus). The antibodies used are listed in Table 1.

### 4.5. RT-qPCR

On days 7 and 14 of the differentiation process, the EBs were carefully isolated using wide-mouthed pipette tips. Cultured EBs (ES-J1-EB) were collected as a negative control, and differentiated EBs (on days 7 and 14) were pooled and collected as a positive control. To analyze the expression of genes in the differentiated EBs, 1 µg/µL of RNA was isolated from the cells of five or more EBs using the GeneAll Hybrid-RTM kit (GeneAll Biotechnology, Songpa-gu, Korea) followed by cDNA synthesis (Hyperscript™ Master mix; GeneAll). The expression of otic differentiation markers was determined by qRT-PCR (7500 real-time PCR system; Applied Biosystems, Foster City, CA, USA) using the SYBR^®^ Green PCR kit (Qiagen, Hilden, Germany). The forward and reverse primers (Oligomer, Bioneer, Daejeon, Korea) used are listed in Table 2. Three independent replicate experiments were carried out for each sample.

### 4.6. Statistical Analysis

All data from the experimental and control samples are expressed as means ± standard deviations. At least three replicates per group were used in all experiments, unless stated otherwise. All data were analyzed using GraphPad Prism (version 5; GraphPad Software Inc., La Jolla, CA, USA) or SPSS (version 22; IBM Corp., Armonk, NY, USA) software. The Kolmogorov–Smirnov test was used to determine whether the data were parametric or nonparametric. Significant differences between the control and treatment groups were analyzed using the *t*-test and Mann–Whitney U test for parametric and nonparametric variables, respectively. One-way ANOVA with Dunnett’s test was used to analyze mRNA expression levels, which are expressed as means ±SD. *p*-values less than 0.05 were considered statistically significant.

## 5. Conclusions

We examined the different cell fates of two morphologies and the effects of a Wnt agonist applied during the differentiation process. Increased hair cell-like cell generation and ‘beating’ organoids, which showed spontaneous movement, appeared in the Wnt group. Therefore, refining the time points and limiting the delivery of the agonist into a specific part of the EB are necessary for optimization of otic differentiation enhancement.

## Figures and Tables

**Figure 1 ijms-22-10306-f001:**
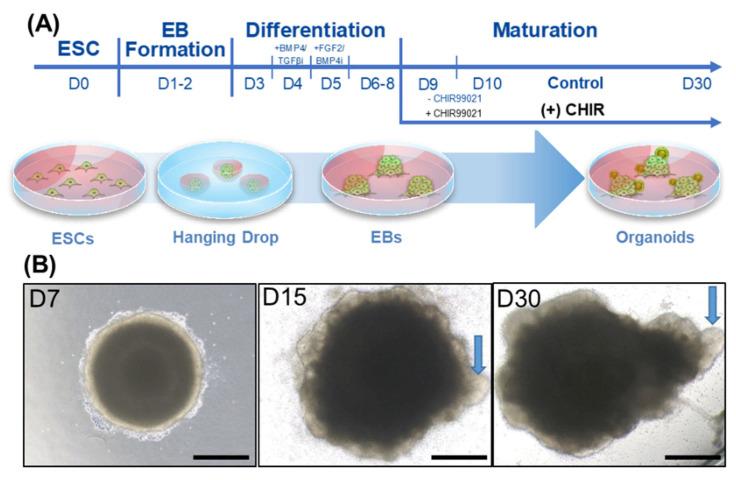
Developmental process of organoids, including hair cell-like cells. (**A**) Generating organoids from mouse embryonic stem cells (ESCs). Hanging drop was used to form cell aggregates or embryonic bodies (EBs). During the differentiation phase, the EBs were placed in differentiation-permissive media followed by the addition of factors that facilitate otic-lineage differentiation (BMP4, TGF-beta inhibitor, FGF2, and BMP4 inhibitors) at specific time points. On day 9, the EBs were transferred into a maturation media with or without the Wnt agonist CHIR99021 for 48 h, and then cultured until day 30. (**B**) A representative image of the maturation phase of (+) CHIR-treated EBs. By day 7 (D7), self-organization of the EB has become apparent. The EBs were composed of a dense central part and a less densely populated peripheral region. On day 15 (D15), multiple outgrowths of cells could be seen in the peripheral region (arrow). On day 30 (D30), these outgrowths had emerged as multiple cystic protrusions (arrow). Scale bars indicate 200 µm.

**Figure 2 ijms-22-10306-f002:**
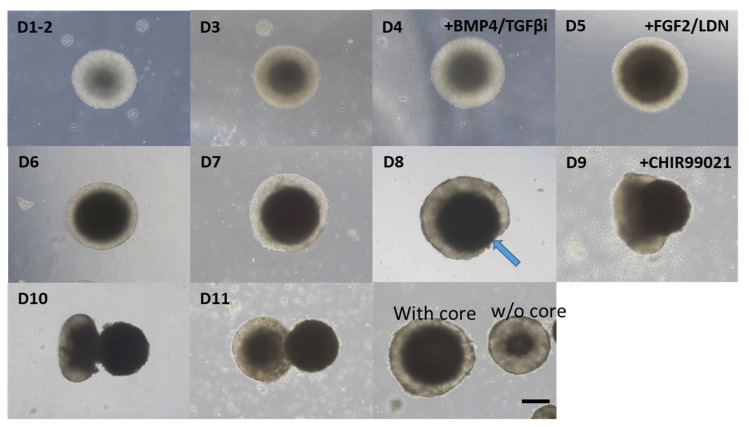
Enucleation of the central core during the maturation phase. Serial changes in a representative EB from day 1 (D1) to day 11 (D11) are shown. During the development of the organoid, the cellular density of the central part of the EB gradually increased, as indicated by the gradual darkening of the central part. Around day 8 (D8), thinning of the peripheral region was observed (arrow). Then, the central part protruded gradually from day 9 (D9) to day 11 (D11). Eventually, this central part (core) detached from the parent EB and formed a separate EB. Some EBs matured without undergoing enucleation. Enucleated EBs were smaller than nonenucleated EBs. Scale bar indicates 200 µm.

**Figure 3 ijms-22-10306-f003:**
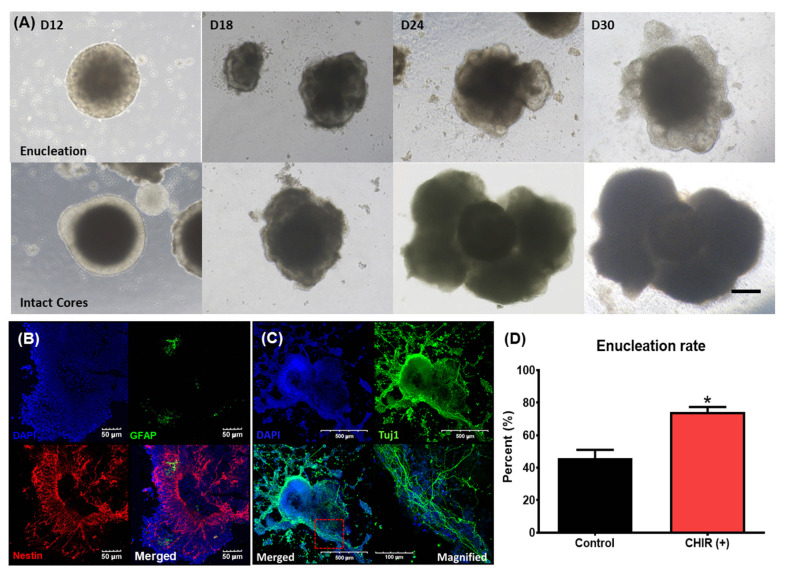
Different fates of EBs (with or without cores) and the effect of a Wnt agonist on enucleation. (**A**) Serial changes in two representative EBs from day 12 (D12) to day 30 (D30). The upper section shows the maturation of EBs after core enucleation. After core enucleation, protrusions around the surface started to develop and form otic organoids. The lower section shows the maturation of EBs with an intact core. These EBs gradually increased in size and developed into cell-rich organoids with multiple lobular projections. The scale bar in image (**A**) indicates 200 µm. Epifluorescence images of fully differentiated EBs (day 30) that did not undergo enucleation are shown in (**B**,**C**). (**B**) The distribution of GFAP- and nestin-positive cells throughout the whole organoid, suggesting that these organoids may have been neural structures. (**C**) Tuj-positive linear structures indicative of nerve fiber sprouting. The magnified image shows multiple linear structures spreading in the same direction. The scale bars in images (**B**,**C**) indicate different lengths. The application of CHIR, which is a Wnt agonist, altered the rate of enucleation. CHIR-treated EBs showed a higher rate of enucleation compared to control (no CHIR) EBs (**D**). (* *p* < 0.05).

**Figure 4 ijms-22-10306-f004:**
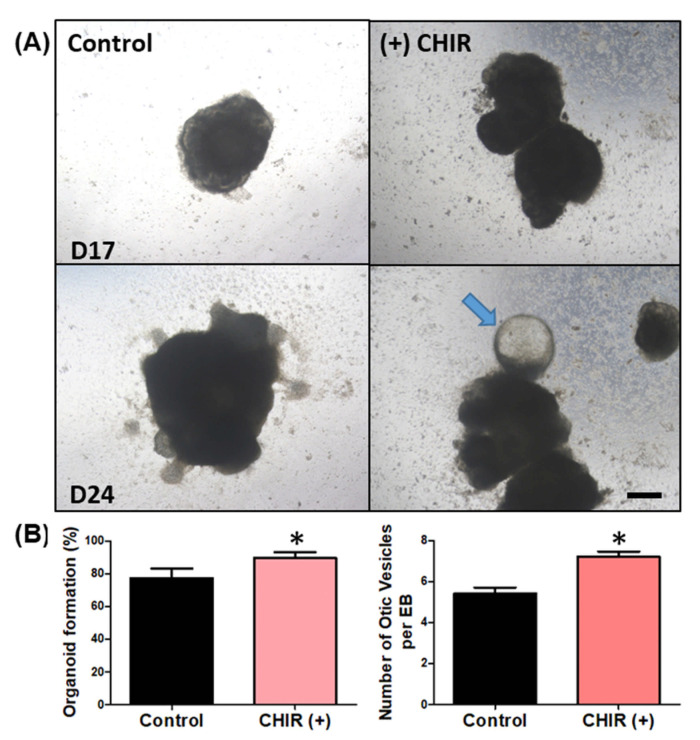
Wnt agonist (CHIR) application and possible otic organoids (cystic protrusions). The images in (**A**) show organoid differentiation at two time points: day 17 (D17) and day 24 (D24). The left column shows organoid differentiation without the Wnt agonist (CHIR), and the right column shows differentiation with CHIR. At D17, the control organoids had irregular surfaces but no definite protrusions, while the CHIR-treated EBs showed clearly protruding lesions around the surface. On D24, irregular surfaces started to develop into protruding lesions in the control organoids, while CHIR-treated EBs showed protrusions that were more enlarged with cystic organoids (arrow), resembling otic organoids. The scale bar in image (**A**) indicates 200 µm. We assessed the organoid formation rate and counted the number of organoids (vesicles) per EB. In the CHIR-treated group, the rate of organoid formation was significantly higher (**B**) and there were more organoids per EB (**C**). (* *p* < 0.05).

**Figure 5 ijms-22-10306-f005:**
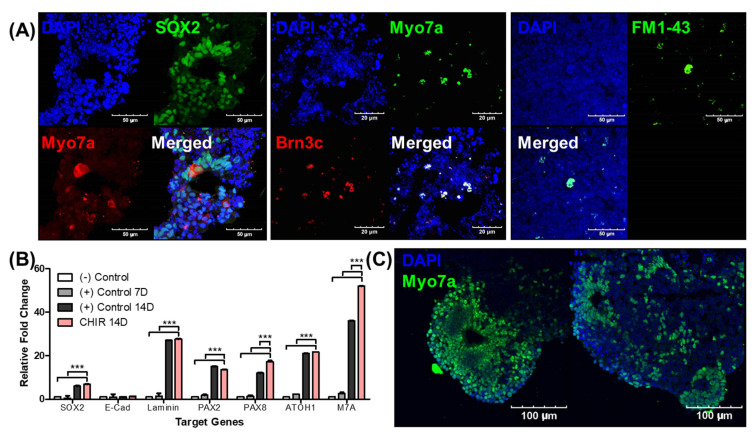
Epifluorescence analysis of organoids and the effect of a Wnt agonist (CHIR) on myosin VIIa expression. We conducted epifluorescence analysis of the organoids from enucleated EBs on differentiation day 30 to measure the hair cell markers, as demonstrated in (**A**). The organoids with mature protrusions were frozen and sectioned. The left image shows myosin VIIa- and Sox2-positive cells in an organoid. Cellular spaces were observed within the organoids. Most of these spaces were surrounded by myosin VIIa (red; inner ear hair cell marker)- and Sox2 (green; inner ear-supporting cell marker)-positive cells. The middle image shows the diffuse expression of myosin VIIa (red)-positive cells and Brn3c (green; another inner ear hair cell marker)-positive cells. Co-expression of myosin VIIa and Brn3c was also observed. The right image shows the expression of FM1-43, which is a marker for mechanoelectrical transduction. (**B**) The results of a RT-PCR analysis of gene expression in accordance with otic differentiation. The negative control ((−) Control) corresponds to EBs before differentiation, (+) Control 7D to EBs on differentiation day 7, (+) Control 14D to EBs on differentiation day 14, and CHIR 14D to EBs with CHIR on differentiation day 14. The graph shows statistically significant differences between CHIR 14D and other time points/conditions. The expression of *Pax8* (otic precursor marker) and *myosin VIIa* (*M7A*) was significantly higher with CHIR compared to (+) Control 14D. The image in (**C**) shows two EBs after 30 days of differentiation with CHIR application. Increased expression of myosin VIIa (green; Myo7a)-positive cells was identified, suggesting enhancement of the otic differentiation process. The scale bars in images (**A**,**C**) indicate different lengths. (*** *p* < 0.001).

**Figure 6 ijms-22-10306-f006:**
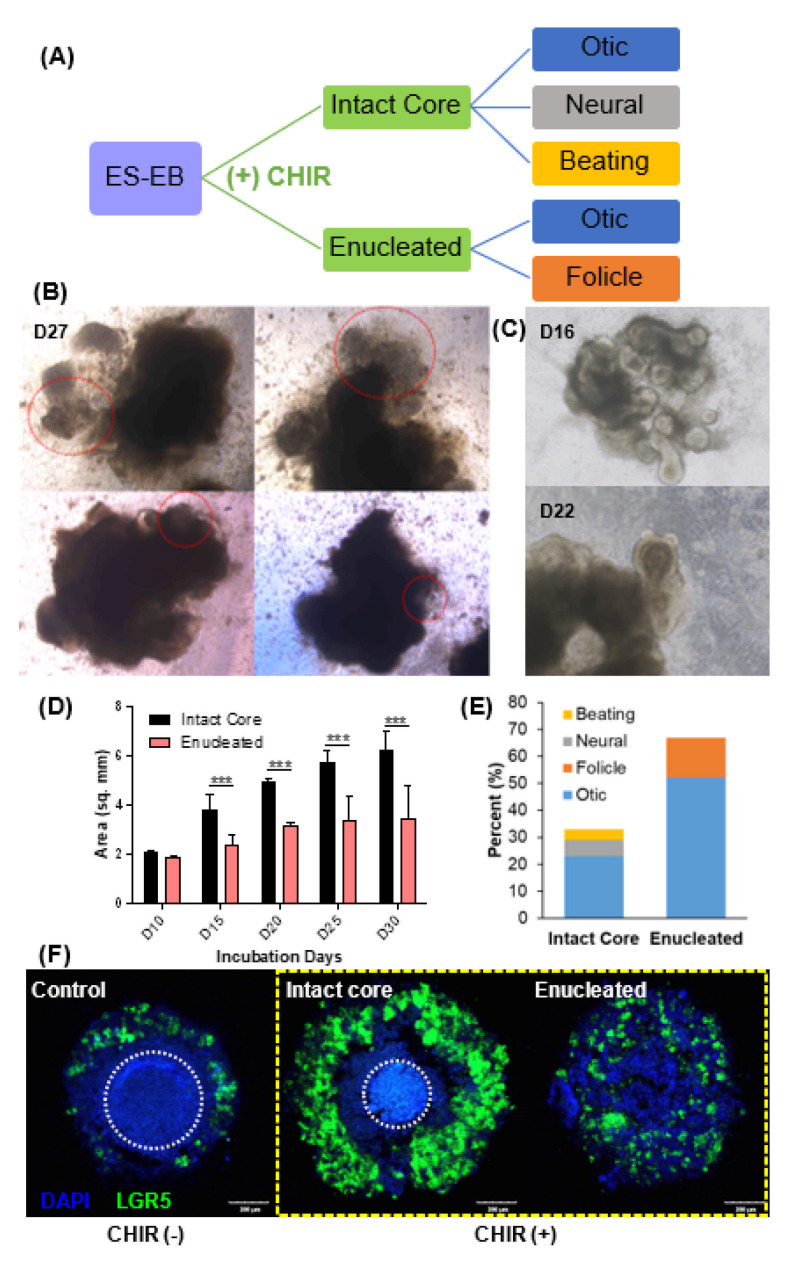
Evidence of differentiation into other lineages after application of a Wnt agonist (CHIR) (observation of ‘beating’ organoid). (**A**) Illustration of the different lineages arising from the core retention or enucleation. CHIR-treated EBs showed atypical morphologies, as shown in (**B**,**C**). On differentiation day 27 (D27) in CHIR-treated EBs, structures with round morphologies (red dotted line) were observed in both EBs with intact cores, as well as enucleated EBs. These structures showed spontaneous movements (‘beating’-like movements; Appendix A) suggesting alternative lineage differentiation (**B**). In addition, enucleated EBs (**C**) also generated hair follicle-like structures on differentiation day 16 (D16) and day 22 (D22). A comparison of the sizes between intact and enucleated EBs showed statistically significant differences starting at day 15 (**D**). All the organoid-bearing EBs after (+) CHIR treatment were classified according to the kind of organoids they developed (**E**). We conducted epifluorescence analysis of LGR5 expression in EBs at D11 after the treatment of CHIR99021 (**F**). The Wnt pathway target LGR5 was expressed in the control group without the addition of CHIR. There was an increase in LGR5 expression after CHIR treatment (yellow dotted rectangle) in EBs with intact cores, as well as in enucleated EBs. Encircled area by white dotted circle indicates the core of control and intact EBs. Scale bar indicates 200 µm. (*** *p* < 0.001).

**Figure 7 ijms-22-10306-f007:**
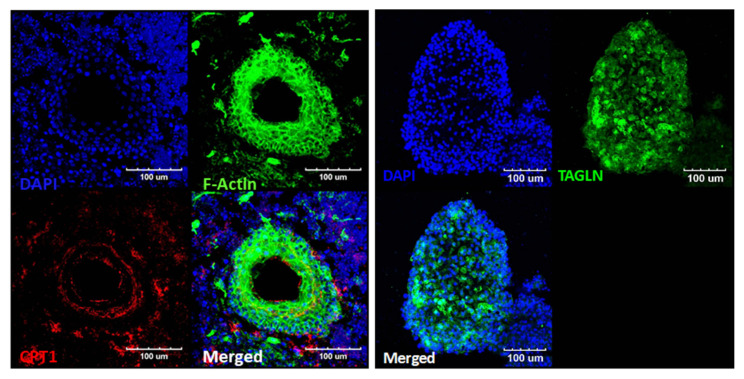
Epifluorescence analysis and incidence of ‘beating’ structures (organoids) (**A**) Results of an epifluorescence analysis of the beating structures. The right image shows the expression of TNNI1 (red; troponin I; slow skeletal muscle gene expressed in cardiomyocytes). The expression of dense F-actin (right image) and TAGLN (left image) (green; transgelin; smooth muscle cell marker) in this structure suggests a muscular composition. Considering the expressed markers, we speculated that these structures were cardiac-mimicking organoids. We compared the number of these beating structures (possible cardiac organoids) between the control (conventional otic differentiation) and CHIR (+) differentiation groups (**B**). No beating organoids were observed in the control group, but they were observed in about 30% of all differentiated organoids in the CHIR-treated group. Scale bars indicate 100 µm.

**Table 1 ijms-22-10306-t001:** Antibodies.

Antibody	Host	Supplier	Catalog No.	Dilution
Brn3c	Mouse	Santa Cruz	Sc81980	1:25
CTPI	Rabbit	Abcam	ab47003	1:100
F-actin (Fluorescein Phalloidin)		Thermo Fisher Scientific	F432	1:100
GFAP	Chicken	Abcam	ab4674	1:1000
MyosinVIIa	Rabbit	Proteus	256790	1:100
Nestin	Mouse	Abcam	ab11306	1:100
SOX2	Mouse	BD Biosciences	561469	1:100
TAGLN (SM22)	Rabbit	Abcam	ab14106	1:200
TuJ1	Mouse	Biolegend	836504	1:100

**Table 2 ijms-22-10306-t002:** Oligomers associated with differentiation marker into inner ear-like structure.

Oligomer	Forward Sequence	Reverse Sequence
*Atoh1*	5′-TCCTATGAAGGAGGTGCGGG-3′	5′-TTAGGGCCCTGTCCTCGAAG-3′
*E-cadherin*	5′-TCTTAGGCACCCAGTAGGCC-3′	5′-TTCCAGGGAGACTGCTAGGC-3′
*GAPDH*	5′-AGGTCGGTGTGAACGGATTTG-3′	5′-GTAGACCATGTAGTTGAGGTCA-3′
*Laminin-β1*	5′-CACCCCTAGCCAACTTGCTG-3′	5′-CTTTGTTCTCCTCACCCGGC-3′
*Myo7a*	5’-CACCAAGGGAGATTGTGGCC-3´	5´-CCTTGGACACCATGACACGG-3´
*Pax2*	5′-GACAGCACCAGACAAGAGGC-3′	5′-TAGCCAAAAAGCCTCGGCAG-3′
*Pax8*	5′-CTTTGCAGTCCCCAGCTCAG-3′	5′-GCCAAGTGCTCTCCTGTGTC-3′
*Sox2*	5´-CACCCCTAGCCAACTTGCTG-3´	5´-CTTTGTTCTCCTCACCCGGC-3´

## Data Availability

The data that support the findings of this study are available from the corresponding author upon reasonable request.

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
