# Peer review of "Wnt Modulation Enhances Otic Differentiation by Facilitating the Enucleation Process but Develops Unnecessary Cardiac Structures"

_ijms, 2021, doi:10.3390/ijms221910306_

Round 1

Reviewer 1 Report

The manuscript by Nathaniel T. Carpena et al examines the role of the WNT pathway in the regulation of otic organoid differentiation by promoting the process of ‘enucleation’. Overall this manuscript is very well written and the data is presented quite convincingly. However, a few points require additional attention.

  1. In figure 3D, there is no error bar, so the author did not repeat the experiment?

  1. In figure 7B, there should have percent (%)

  1. Though the author revealed that the non-enucleated EBs differentiated into neural lineage, the author should directly confirm the ‘enucleation’ organoid could differentiate into otic organoid by immunofluorescence with some otic special markers.

  1. In figure 5B, the author should add some WNT pathway targets to make sure the WNT agonist (Chir) works well in the experiments

  1. If the author can examine the WNT pathway activity in ‘enucleation’ organoid and non-enucleated organoid?

Author Response

We would like to thank the reviewers for their comments and suggestions. We have revised our manuscript accordingly and added new data as needed. Reviewers’ comments are in bold letters and answers to each follows the mark “ANS.” Changes within the manuscript were tracked using the “Track Changes” feature of MS Word as advised by the editor

Reviewer 1

The manuscript by Nathaniel T. Carpena et al examines the role of the WNT pathway in the regulation of otic organoid differentiation by promoting the process of ‘enucleation’. Overall this manuscript is very well written and the data is presented quite convincingly. However, a few points require additional attention.

In figure 3D, there is no error bar, so the author did not repeat the experiment?

ANS. The experiment was replicated thrice. We have added the appropriate statistical notations on Figure 3D.

In figure 7B, there should have percent (%)

 ANS. We removed this figure and added the data to Fig6C for better presentation our findings.

Though the author revealed that the non-enucleated EBs differentiated into neural lineage, the author should directly confirm the ‘enucleation’ organoid could differentiate into otic organoid by immunofluorescence with some otic special markers.

 ANS. We have now clarified that the samples used in Figure 5 to confirm the presence of otic organoids are from enucleated EBs.

Line 186. “In the final stage of the differentiation process, enucleated EBs with otic vesicles and suspected inner ear organoids were fixed and prepared for epifluorescence analysis and real-time PCR."

Line 212. We conducted epifluorescence analysis of the organoids from enucleated EBs on differentiation day 30…”

In figure 5B, the author should add some WNT pathway targets to make sure the WNT agonist (Chir) works well in the experiments

ANS. We are not able to conduct another run of RT-PCR during the given time for the revision. However, we have included in Fig6F some additional staining showing the expression of LGR5 in control as well as CHIR treated EBs with and without their cores.

Line 254. “The difference in effect of wnt activation was also compared by observing the expression of the wnt target protein LGR5 at the start of maturation (D12) and found that the expression is highly elevated in CHIR treated samples (Fig. 6F). LGR5 expression can be observed in the control (-) CHIR group but CHIR treated EBs showed higher LGR5 expression. Intact EBs showed high expression of LGR5 only in the outer layer. Cells expressing LGR5 became evenly distributed within the EB after enucleation.”

If the author can examine the WNT pathway activity in ‘enucleation’ organoid and non-enucleated organoid?

ANS. The examination of the whole molecular pathway involving the divergent effect of enucleation on the differentiation of ESCs following the Wnt pathway is beyond the scope of this study. We will be conducting a more detailed study on the matter following this article.

Reviewer 2 Report

Line 20 – Sensory hair cell-like sounds better for me here in the abstract: “increased hair cell-like cell generation”

Line 20 – Please explain why “with a longer incubation period” if the author didn’t compare distinct treatment time

Line 65 – The authors used a protocol adapted from Koehler et al. and also DeJonge et al. since the latter introduced the use of CHIR99021.

Line 67 – Call Fig1A

Line 67 – This sentence is unnecessary “Prior to the actual developmental process, mouse ESCs were cultivated in the stem cell media to inhibit the differentiation of pluripotent cells.”

Line 69-72 – Hang drop description should be at Material and methods section. Is it the same as ref 12? If so, itshould be cited here and/or M&M

Line 73 – When CHIR99021 was added? Day 9 or day 7? For how long? Please clarify here and at M&M

Line 74 – Should be mentioned here that there were two different groups: with and without CHIR99021 during maturation (or control and treated). This could also be shown in the Fig1A, for instance with two distinct branches in maturation (-) and (+) CHIR99021.

Line 75 and 78 – Call Fig1B

Fig1A could be smaller. Figure legend could be summarized. Fig1B shows EBs treated with CHIR99021 or not? Main text said that but legend not, please clarify.

Line 90 – “By day 7 (D7), the EBs had appeared”. EBs appear by day 1-2, please correct that.

Line 127 – How about the neuronal markers in enucleated EBs. I’d be interesting show them as a control, maybe even at supplementary. Are there more replicates for GFAP staining? How about the negative control for the staining? I’d say that it is hard to distinguish from background. Maybe add a zoomed image.

Fig2 and 3 – How many experimental replicates were used? Quantification of enucleation rate should have an error bar and statistic test.

Fig2 and 3A, B – EBs were treated with CHIR99021? I would highly recommend show similar results using another pluripotent/embryonic cell line.

Page 5, Fig4 – If now the authors start describing Wnt agonist effects, all they have shown before was without CHIR99021? This is very confusing throughout the results. Fig 4 is called before Fig3D, it should be in order of results description. Fig 4 doesn’t show any new information as already previously described by DeJonge et al.

Line 146 and Fig4 – Please clarify between organoid formation, otic vesicle formation rate and number of otic vesicles per EBs. What would be no organoid formation? Are the results shown from enucleated EBs? Non-enucleated EBs don’t form otic vesicles at all? How is the percentage of enucleated EBs that form otic vesicle?

Line 159 – There isn’t any arrow at the image.

Fig5A – I’m sorry but this figure is awful. The image quality is really bad and I can’t see any reasonable staining.

Line 179-188 – I didn’t understand why the authors are evaluating EBs not treated with CHIR99021 if they have already confirmed that treating is better. All these have been shown by DeLonge et al., there is no reason to keep struggling on that. The authors have some interesting results about the enucleation process – this should be the focus, characterizing this not yet described process.

Line 184 – What the authors intend to say with the sentence: “These results demonstrate a high probability of inner ear organoid differentiation using a previously reported protocol.[4]”

Table 1 is unnecessary. The information is at the gaph 5B. Perhaps go to supplementary

Line 210 – Where is the evidence showed in Fig6A of other lineages differentiation? The illustration only shows a hypothesis of that. In Fig6B, C and Fig 7 the authors show some kind of evidence of other lineages derivation.

Fig6A – At least half of this illustration is very similar to Fig1 from Koehler et al. I’d highly recommend remake it or at least cite in the figure legend as “modified from Koehler et al.” Please explain in the legend the meaning of each abbreviation in the illustration.

Fig6B – Beating organoids also have otic vesicles? Or they are exclusives? The same for hair follicle organoids. Are the beating and hair follicle EBs derived from enucleated EBs? What is the reference for “previously reported hair organoids”?

Fig7 – Would the quantification of organoid formation in Fig4B include the “other lineages” EBs?

Fig7B – Please add a name for the y axis. There is no * in the graph. This graph could be smaller and occupy the black square bellow TAGLN staining, that doesn’t mean anything.

Discussion

Line 253 – In the sentence: “Enucleated EBs were similar to inner ear organoids, as demonstrated in previous reports.[12]”, please explain why cite ref 12 here. Ref 12 doesn’t use CHIR99021 and the authors here did not convince me yet that the enucleated EBs are similar to inner ear organoids.

How much these other lineages EBs interfere with the final outcome of the culture? For instance, adding CHIR99021 the outcome is improved, on the other hand you can have beating EBs. But without CHIR99021 the otic outcome is lower, and no beating EBs are seen. What is more important here? A good outcome or specificity? My point is, even if you get some beating EBs you also get more otic EBs. And if you don’t get beating EBs, you get something unknown and not otic.

Is there some kind of selection for forward use? For further studies and potential use of otic differentiated cells, the EBs that show otic vesicles are selected, right?

Line 265 – In the sentence: “One of the novel findings of the current study was the differentiation into alternative lineages, as indicated by the presence of beating movements in what appear to be cardiac organoids.” Has anyone before studied the non-otic EBs derived in the protocols to look for other lineages? It doesn’t seen odd to me find many other lineages since otic vesicles are very delicate and specific structures. What is the overall percentage of EBs with otic vesicles?

I really missed more characterization of enucleated EBs vs non enucleated EBs and how this would help further studies in the field. Does the enucleation improve otic vesicles outcome? This is not clear.

Line 284 – “The density and total number of cells in EBs or aggregates may regulate enucleation.” Maybe the author could test this and also see if enucleation varies between different cell lines.

It’d interesting if the discussion includes some comments regarding why Wnt could facilitate enucleation and that’s why the outcome is better. I also wonder what happen with the eject core. Is there a way to trace that?

Materials and Methods

It should be mentioned here: protocol adapted from Koehler et al. and DeJonge et al.

Why use hanging drop technique?

Non-neural ectodermal differentiation medium described in line 339 is the same as the one cited in line 353? And it would be the basal medium cited in line 354, in which inducing factors were added? - Please clarify

I’d suggest clarify that SB431542 (Stemgent) and LDN-193189 (Stemgent) are TGFbi and BMP4i mentioned in Fig1A.

Why start CHIR at day 9 when DeJonge showed the best day way day8-10? Was it adapted for this cell line?

Table 2:

GFAP ab - ab4764 doesn’t correspond to GFAP. Please correct.

Tuj1 ab – 801201 ref num is not from Santa Cruz. Please correct and review all ab.

Author Response

We would like to thank the reviewers for their comments and suggestions. We have revised our manuscript accordingly and added new data as needed. Reviewers’ comments are in bold letters and answers to each follows the mark “ANS.” Changes within the manuscript were tracked using the “Track Changes” feature of MS Word as advised by the editor (Please see the attached file with answers with figure).

Reviewer 2

  1. Line 20 – Sensory hair cell-like sounds better for me here in the abstract: “increased hair cell-like cell generation”

ANS. We agree with the suggestion of the reviewer and revised our manuscript accordingly.

Line 20. “…leading to sensory hair cell-like cell generation.”

  1. Line 20 – Please explain why “with a longer incubation period” if the author didn’t compare distinct treatment time

ANS. Previous published papers showing the differentiation of mouse ESCs were only from 14 to 21 days, we observed the immergence of the beating organoids after 21 days of culture.

  1. Line 65 – The authors used a protocol adapted from Koehler et al. and also DeJonge et al. since the latter introduced the use of CHIR99021.

ANS. We have added DeJonge in citing the protocol that we followed.

  1. Line 67 – Call Fig1A

ANS. We have corrected the proper figure citation.

  1. Line 67 – This sentence is unnecessary “Prior to the actual developmental process, mouse ESCs were cultivated in the stem cell media to inhibit the differentiation of pluripotent cells.”

ANS. We agree with the reviewer and omitted this sentence.

  1. Line 69-72 – Hang drop description should be at Material and methods section. Is it the same as ref 12? If so, it should be cited here and/or M&M

ANS. We removed the details of the hanging drop method in the Results and added the proper citation here as well as in the M&M.

  1. Line 73 – When CHIR99021 was added? Day 9 or day 7? For how long? Please clarify here and at M&M

ANS. We clarified the schedule of CHIR99021 treatment as follows:

Line 427. “The treated cells were cultured for 3 days, and the medium was replaced on day 79 with maturation medium containing 1% Matrigel with or without 3 µM CHIR99021. After 48 hrs, a complete change of maturation media without Matrigel and CHIR99021 was done. Half of the medium was replaced with maturation medium without Mat-rigel and CHIR99021 every other day until day 30.”

  1. Line 74 – Should be mentioned here that there were two different groups: with and without CHIR99021 during maturation (or control and treated). This could also be shown in the Fig1A, for instance with two distinct branches in maturation (-) and (+) CHIR99021.

ANS. We have made the necessary changes in Fig1A and in the text as follows:

Line 74. “At day 9, the EBs were transferred to maturation media including with or without the Wnt ag-onist CHIR99021 (control and (+) CHIR),…”

  1. Line 75 and 78 – Call Fig1B

ANS. We have modified the figure citation as follows;

Line 76. “As shown in Figure 1B,…”

  1. Fig1A could be smaller. Figure legend could be summarized. Fig1B shows EBs treated with CHIR99021 or not? Main text said that but legend not, please clarify.

ANS. Figure 1 was reduced in size and the figure legend was edited as follows;

Line 84. “Image (A) illustrates the process of generating organoids from mouse embryonic stem cells (ESCs). Hanging drop was used to form cell aggregates or embryonic bodies (EBs). During the differentiation phase, the EBs are placed in differentiation-permissive media followed by the addition of factors that facilitate otic-lineage differentiation (BMP4, TGF-beta inhibitor, FGF2, and BMP4 inhibitors) at specific time points. On day 9, the EBs were transferred into a maturation media with or without the Wnt agonist CHIR99021 for 48 h then cultured until day 30. Image (B) shows a representative image of the maturation phase of (+) CHIR treated EBs.”

  1. Line 90 – “By day 7 (D7), the EBs had appeared”. EBs appear by day 1-2, please correct that.

ANS. We have changed the description for first image of Fig1B.

Line 98. “By day 7 (D7), self-organization of the EB has become apparent.”

  1. Line 127 – How about the neuronal markers in enucleated EBs. I’d be interesting show them as a control, maybe even at supplementary. Are there more replicates for GFAP staining? How about the negative control for the staining? I’d say that it is hard to distinguish from background. Maybe add a zoomed image.

ANS. We replaced the GFAP staining image with a higher magnification image. No GFAP-positive cells were found in enucleated EBs.

  1. Fig2 and 3 – How many experimental replicates were used? Quantification of enucleation rate should have an error bar and statistic test.

ANS. Three experimental replicates were conducted for the study and Fig2 shows the representative data. The appropriate statistical information was added to Fig3D.

  1. Fig2 and 3A, B – EBs were treated with CHIR99021? I would highly recommend show similar results using another pluripotent/embryonic cell line.

ANS. We are not able to test whether the addition of CHIR99021 also show the same enucleation process in other cell lines during differentiation. We have added this in the discussion as a limitation of our study and recommended this for a future study.

Line 345. “We also recommend the use of other embryonic or pluripotent stem cell line to investigate if the similar enucleation process occurs upon the modulation of the Wnt signaling pathway or perhaps may only be limited to the cell line used in this study.”

  1. Page 5, Fig4 – If now the authors start describing Wnt agonist effects, all they have shown before was without CHIR99021? This is very confusing throughout the results. Fig 4 is called before Fig3D, it should be in order of results description. Fig 4 doesn’t show any new information as already previously described by DeJonge et al.

ANS. We transferred the paragraph below from line 153 to 176 to avoid make the flow of results clearer. Figure 4 shows the positive results of the enucleated EBs upon the application of CHIR as shown previously by DeJonge et al.

Line 176. “Previous reports have indicated that modulating/enhancing the Wnt pathway could im-prove the differentiation of otic organoids. Therefore, we attempted to modulate the differentiation process via application of a Wnt agonist (CHIR99021). Indeed, Wnt agonist appli-cation increased both otic vesicle formation rate and number of otic vesicles per EBs (Fig. 4).”

  1. Line 146 and Fig4 – Please clarify between organoid formation, otic vesicle formation rate and number of otic vesicles per EBs. What would be no organoid formation? Are the results shown from enucleated EBs? Non-enucleated EBs don’t form otic vesicles at all? How is the percentage of enucleated EBs that form otic vesicle?

ANS. We added the following to clarify these matter;

Line 179. ”The rate of organoid formation indicates the number chances of developing any organoid structure, whether otic, neural, beating or hair, from all the generated EBs from the hanging drop. EBs that do not form any organoid do not show any changes in morphology and simply break apart or detach from the culture plate. The otic vesicle formation rate indicates how many otic vesicles among the developed organoid structure are. The number of otic vesicles per EB was counted depending on how many protruding otic organoids are produced in from a single EB.”

Line 250. “In Figure 6E, 75% formed otic vesicle out of all the EBs that developed organoids 25% of which came from intact EBs while 52% are from enucleated EBs.”

  1. Line 159 – There isn’t any arrow at the image.

ANS. We added the arrow pointing at the otic organoid in Fig4A.

  1. Fig5A – I’m sorry but this figure is awful. The image quality is really bad and I can’t see any reasonable staining.

ANS. We are very sorry the image came out that way and properly adjusted the images.

  1. Line 179-188 – I didn’t understand why the authors are evaluating EBs not treated with CHIR99021 if they have already confirmed that treating is better. All these have been shown by DeLonge et al., there is no reason to keep struggling on that. The authors have some interesting results about the enucleation process – this should be the focus, characterizing this not yet described process.

ANS. We included these data in order to show duplicate results of the Wnt modulation in generating otic organoids using another stem cell line. We have included further characterization of the EBs with and without cores in Fig6.

Line 246. “The difference between intact and enucleated EBs was also apparent several days after CHIR treatment. There is a statistically significant mean size difference of 2.10 ± 0.58 mm2 between the intact and enucleated EBs at day 15 until the end of the experiment at day 30 (Fig.6D). The difference in EB size is largely affected by the enucleation process as shown by the disappearance of the core that can be found in the control and intact EBs (red encircled area). In Figure 6E, 75% formed otic vesicle out of all the EBs that developed organoids 25% of which came from intact EBs while 52% are from enucleated EBs. Intact EBs also generated 6% neural and 4% beating organoids, none of which appeared in enucleated EBs. Organoids resembling hair follicles appeared from enucleated EBs which represents 15% of the total EBs observed. The difference in effect of Wnt activation was also compared by observing the expression of the Wnt target protein LGR5 at the start of maturation (D12) and found that the expression is highly elevated in CHIR treated samples (Fig. 6F). LGR5 expression can be observed in the control (-) CHIR group but CHIR treated EBs showed higher LGR5 expression. Intact EBs showed high expression of LGR5 only in the outer layer. Cells expressing LGR5 became evenly distributed within the EB after enucleation.”

  1. Line 184 – What the authors intend to say with the sentence: “These results demonstrate a high probability of inner ear organoid differentiation using a previously reported protocol.[4]”

ANS. We deleted this sentence in this revision.

  1. Table 1 is unnecessary. The information is at the graph 5B. Perhaps go to supplementary

ANS. We moved Table 1 as a supplementary data.

  1. Line 210 – Where is the evidence showed in Fig6A of other lineages differentiation? The illustration only shows a hypothesis of that. In Fig6B, C and Fig 7 the authors show some kind of evidence of other lineages derivation.

ANS. We decided to remove Fig6A because we haven’t found a clear pathway to illustrate the emergence of the other differentiated organoids.

  1. Fig6A – At least half of this illustration is very similar to Fig1 from Koehler et al. I’d highly recommend remake it or at least cite in the figure legend as “modified from Koehler et al.” Please explain in the legend the meaning of each abbreviation in the illustration.

ANS. We decided to remove this image since we did not investigate the distinct pathways that led to the generation of the other organoid lineages.

  1. Fig6B – Beating organoids also have otic vesicles? Or they are exclusives? The same for hair follicle organoids. Are the beating and hair follicle EBs derived from enucleated EBs? What is the reference for “previously reported hair organoids”?

ANS. We have added Fig6E in order to clearly explain the origins of the different organoids. We transferred the sentence containing “previously reported hair organoids” into Line 234 and added the appropriate citation.

Line 250. “In Figure 6E, 75% formed otic vesicle out of all the EBs that developed organoids 25% of which came from intact EBs while 52% are from enucleated EBs. Intact EBs also generated 6% neural and 4% beating organoids, none of which appeared in enucleated EBs. Organoids resembling hair follicles appeared from enucleated EBs which represents 15% of the total EBs observed.”

  1. Fig7 – Would the quantification of organoid formation in Fig4B include the “other lineages” EBs?

ANS. The other lineages are included in counting the total EBs that formed organoids.

Line 179. “The rate of organoid formation indicates the number chances of developing any organoid structure, whether otic, neural, beating or hair, from all the generated EBs from the hanging drop.”

  1. Fig7B – Please add a name for the y axis. There is no * in the graph. This graph could be smaller and occupy the black square bellow TAGLN staining, that doesn’t mean anything.

ANS. The graph in Fig7B was removed and the data was included in Fig6C.

Discussion

  1. Line 253 – In the sentence: “Enucleated EBs were similar to inner ear organoids, as demonstrated in previous reports.[12]”, please explain why cite ref 12 here. Ref 12 doesn’t use CHIR99021 and the authors here did not convince me yet that the enucleated EBs are similar to inner ear organoids.

ANS. We cited ref 12 here in a limited instance that the inner ear organoids that we got with the use of CHIR resembles the results thus providing a partial identification of the existence otic organoids. We are hesitant to cite the DeJonge paper because our hair cells are not as mature thus leading us to use the term “inner ear hair-cell-like cells”.

  1. How much these other lineages EBs interfere with the final outcome of the culture? For instance, adding CHIR99021 the outcome is improved, on the other hand you can have beating EBs. But without CHIR99021 the otic outcome is lower, and no beating EBs are seen. What is more important here? A good outcome or specificity? My point is, even if you get some beating EBs you also get more otic EBs. And if you don’t get beating EBs, you get something unknown and not otic.

ANS. We have added the following paragraph in the discussion;

Line 379. “Organoids has provided an opportunity to recreate the structure and physiology of organs. Likewise, diseases and treatments are also possible to investigate using or-ganoids. Our study has shown that incorrect implementation of a certain treatment can also cause undesirable effects. Compared to the endogenous control of Wnt regula-tion, the bulk treatment of the Wnt agonist CHIR99021 on an EB That Wnt agonist ap-plication also led to differentiation of other lineages. This indicates that a more target-ed delivery of Wnt agonists is necessary to enhance the otic differentiation process. Re-fining the time points and limiting the delivery of the agonist into a specific part of the EB, such as a putative progenitor area, could be a good approach. Furthermore, the in-ner ear could benefit more from a more informed effect of stimulating factors that are used if we ever hope to achieve the intricate architecture of the inner ear.”

  1. Is there some kind of selection for forward use? For further studies and potential use of otic differentiated cells, the EBs that show otic vesicles are selected, right?

ANS. If we ever hope to apply the EBs for potential future studies the presence of otic vesicles will surely be a selective measure. However, for example, we have noticed that for implantation of differentiated stem cells that partially differentiated stem cells fare better when implanted in vivo. That is why a full-proof initiation of otic sensory cells is needed to ensure that any implanted cell or organoid will develop only into the desired target cell lineage.

  1. Line 265 – In the sentence: “One of the novel findings of the current study was the differentiation into alternative lineages, as indicated by the presence of beating movements in what appear to be cardiac organoids.” Has anyone before studied the non-otic EBs derived in the protocols to look for other lineages? It doesn’t seen odd to me find many other lineages since otic vesicles are very delicate and specific structures. What is the overall percentage of EBs with otic vesicles?

ANS. We have added the overall chances of occurrence for the different lineages in Fig.6C. Adding together the otic vesicle forming EBs with intact cores and enucleated EB gets a total of 75%.

  1. I really missed more characterization of enucleated EBs vs non enucleated EBs and how this would help further studies in the field. Does the enucleation improve otic vesicles outcome? This is not clear.

ANS. We added these sentences to note the implications of retaining or ejecting the cores;

Line 333. “This would affect the overall fate of the EB, as it would eventually differentiate into a neural organoid that could, in theory, provide a source of neural connections for the developing otic organoids. On the other hand, the enucleation removes the largest source of undifferentiated pluripotent cells that could develop into other lineages.”

  1. Line 284 – “The density and total number of cells in EBs or aggregates may regulate enucleation.” Maybe the author could test this and also see if enucleation varies between different cell lines.

ANS. We included this in the discussion as another probable cause of enucleation since the cell density at the formation of EBs is considerably higher compared to other published paper. And indeed this can be looked into for further investigation of the enucleation phenomena.

  1. It’d interesting if the discussion includes some comments regarding why Wnt could facilitate enucleation and that’s why the outcome is better. I also wonder what happen with the eject core. Is there a way to trace that?

ANS. The ejected core progressively dissociated and washed away during media changes as mentioned in line 112.  We added the following paragraph to comment on how Wnt could possibly facilitate enuclation;

Line 349. “ In looking at the relationship of Wnt signaling and enucleation, a similar occur-rence of tissue patterning during development of the embryo happens during gastrulation. Wnt signaling activity coordinates the action of signals during the progression of gastrulation [15]. Ligands from Wnt producing cells regulates the signaling activity at gastrulation. A mutation on these Wnt ligands fail to initiate gastrulation and the epi-blast persistently expresses pluripotency and neural progenitor markers of the embryos.”

Materials and Methods

  1. It should be mentioned here: protocol adapted from Koehler et al. and DeJonge et al.

ANS. We have added both citation.

  1. Why use hanging drop technique?

ANS. Hanging drop remains a simple and reliable way of generating EBs without the use of special culture plates.

  1. Non-neural ectodermal differentiation medium described in line 339 is the same as the one cited in line 353? And it would be the basal medium cited in line 354, in which inducing factors were added? - Please clarify

ANS. Yes, these use the same ectodermal differentiation media as a base. We modified the text to remove the confusion;

Line 408. “Ectodermal differentiation was initiated using Glasgow minimum essential medium…”

  1. I’d suggest clarify that SB431542 (Stemgent) and LDN-193189 (Stemgent) are TGFbi and BMP4i mentioned in Fig1A.

ANS. We clarified the identity of the two inhibitors.

Line 390. “Then, on the second day of differentiation, non-neural ectoderm was induced by add-ing 10 ng/mL recombinant BMP4 (Stemgent, Beltsville, MD, USA) and 1 μM TGFβi (SB431542, (Stemgent) to the cultured cells. On the third day, preplacodal ectoderm was induced by adding 25 ng/mL FGF-2 (PeproTech, Rocky Hill, NJ, USA) and 1 μM BMP4i (LDN-193189, (Stemgent).”

  1. Why start CHIR at day 9 when DeJonge showed the best day way day8-10? Was it adapted for this cell line?

ANS. The schedule for the CHIR treatment was adapted to accommodate the time used for the hanging drop to generate the EBs.

  1. Table 2: GFAP ab - ab4764 doesn’t correspond to GFAP. Please correct. Tuj1 ab – 801201 ref num is not from Santa Cruz. Please correct and review all ab.

ANS. We updated the correct details for GFAP and Tuj1 antibodies and now designated as Table 1.

Round 2

Reviewer 1 Report

Accept in present form

Reviewer 2 Report

All the improvements helped the manuscript looks much better.

My last suggestion would be to the title: Wnt modulation enhances otic differentiation but develops unnecessary cardiac structures by facilitating the enucleation process - makes me understand that the development of cardiac structures are conditional to the enucleation process, and that is not true by your results. Also, cardiac strutures were the last abundant between the other developed lineages, why highlight this one? So I suggest change to something like: "Wnt modulation enhances otic differentiation by facilitating the enucleation process but develops other unnecessary lineages"